

# PhilDB: the time series database with built-in change logging

Andrew MacDonald

Melbourne Victoria, Australia

## ABSTRACT

PhilDB is an open-source time series database that supports storage of time series datasets that are dynamic; that is, it records updates to existing values in a log as they occur. PhilDB eases loading of data for the user by utilising an intelligent data write method. It preserves existing values during updates and abstracts the update complexity required to achieve logging of data value changes. It implements fast reads to make it practical to select data for analysis. Recent open-source systems have been developed to indefinitely store long-period high-resolution time series data without change logging. Unfortunately, such systems generally require a large initial installation investment before use because they are designed to operate over a cluster of servers to achieve high-performance writing of static data in real time. In essence, they have a 'big data' approach to storage and access. Other open-source projects for handling time series data that avoid the 'big data' approach are also relatively new and are complex or incomplete. None of these systems gracefully handle revision of existing data while tracking values that change. Unlike 'big data' solutions, PhilDB has been designed for single machine deployment on commodity hardware, reducing the barrier to deployment. PhilDB takes a unique approach to meta-data tracking; optional attribute attachment. This facilitates scaling the complexities of storing a wide variety of data. That is, it allows time series data to be loaded as time series instances with minimal initial meta-data, yet additional attributes can be created and attached to differentiate the time series instances when a wider variety of data is needed. PhilDB was written in Python, leveraging existing libraries. While some existing systems come close to meeting the needs PhilDB addresses, none cover all the needs at once. PhilDB was written to fill this gap in existing solutions. This paper explores existing time series database solutions, discusses the motivation for PhilDB, describes the architecture and philosophy of the PhilDB software, and performs an evaluation between InfluxDB, PhilDB, and SciDB.

Corresponding author
Andrew MacDonald,
andrew@maccas.net

## INTRODUCTION

PhilDB was created to store changing time series data, which is of great importance to the scientific community. In hydrology, for example, streamflow discharge can be regularly updated through changes in quality control processes and there is a need to identify when such data has changed. Efficient access to time series information supports effective and thorough analysis. Currently, existing proprietary and open-source database solutions for storing time series fail to provide for effortless scientific analysis. In practice, the steep learning curves, time-consuming set up procedures, and slow read/write processes are

considerable barriers to using these systems. More critically, most fail to provide the ability to store any changes to a time series over time. Most current open-source database systems are designed for handling 'big data,' which in turn requires extreme computing power on a cluster of servers.

This paper will explore existing time series database solutions. It will examine the need for a liberally licensed, open-source, easily deployed time series database, that is capable of tracking data changes, and look at why the existing systems that were surveyed failed to meet these requirements. This paper will then describe the architecture and features of the new system, PhilDB, that was designed to meet these outlined needs. Finally, an evaluation will be performed to compare PhilDB to the most promising alternatives of the existing open-source systems.

## BACKGROUND: EXISTING SYSTEMS

### Proprietary systems

There are a number of proprietary solutions for storage of time series data that have been around since the mid-nineties to the early 2000s. *Castillejos (2006)* identified three proprietary systems of note, FAME, TimeIQ, and DBank, that have references that range from 1995 to 2000. There are other proprietary systems, such as kdb+ (http://kx.com/software.php), that are commercially available today. This shows that time series data storage is an existing problem. Compared to proprietary systems, open-source systems can generally be used with the scientific Python ecosystem as described by *Perez, Granger & Hunter (2011)*. Ready access to open-source systems also make them easier to evaluate and integrate with. Therefore existing proprietary systems were not evaluated any further. Discussion on the need for an open-source system is further covered in the 'Motivation' section.

### Open-source systems

In recent years the development of open-source time series databases has taken off, with most development beginning within the last five years. This can be seen by the number of projects discussed here along with noting the initial commit dates.

#### *'Big data' time series databases*

Some of the most successful projects in the open-source time series database space are OpenTSDB,[1] Druid,[2] Kairosdb,[3] and InfluxDB.[4] The earliest start to development on these systems was for OpenTSDB with an initial commit in April 2010. These systems are designed to operate over a cluster of servers to achieve high-performance writing of static data in real time. In essence, they have a 'big data' approach to storage and access. The architectural approach to address big data requirements means a large initial installation investment before use.

#### *Alternate time series databases*

In contrast to the 'big data' time series systems some small dedicated open-source code bases are attempting to address the need for local or single server time series data storage. These systems, however, have stalled in development, are poorly documented, or require a

[1] OpenTSDB initial commit: 2010-04-11; https://github.com/OpenTSDB/opentsdb.

[2] Druid initial commit: 2012-10-24; https://github.com/druid-io/druid/.

[3] Kairosdb initial commit: 2013-02-06; https://github.com/kairosdb/kairosdb.

[4] InfluxDB initial commit: 2013-04-12; https://github.com/influxdb/influxdb.

[5]Timestore http://www.mike-stirling.
com/redmine/projects/timestore;
https://github.com/mikestir/timestore
initial commit 2012-12-27.

[6]tsdb initial commit: 2013-01-11; most
recent commit at time of writing: 2013-02-
17; https://github.com/gar1t/tsdb.

[7]Cube initial commit: 2011-09-13;
https://github.com/square/cube.

moderate investment of time to operate. For example Timestore[5] was, at the time of writing, last modified August 2013 with a total development history of 36 commits. Some of the better progressed projects still only had minimal development before progress had ceased; for example, tsdb[6] with a development start in January 2013 and the most recent commit at time of writing in February 2013 for a total of 58 commits. Cube[7] has a reasonable feature set and has had more development effort invested than the other systems discussed here, with a total of 169 commits, but it is no longer under active development according the Readme file. Searching GitHub for 'tsdb' reveals a large number of projects named 'tsdb' or similar. The most popular of these projects (when ranked by stars or number of forks) relate to the 'big data' systems described earlier (in particular, OpenTSDB, InfluxDB, and KairosDB). There are numerous small attempts at solving time series storage in simpler systems that fall short of a complete solutions. Of the systems discussed here, only Cube had reasonable documentation, Timestore had usable documentation, and tsdb had no clear documentation.

### Scientific time series databases

At present, the only open-source solution that addresses the scientific need to track changes to stored time series data as a central principle is SciDB (*Stonebraker et al., 2009*; *Stonebraker et al., 2011*). SciDB comes with comprehensive documentation (http://www.paradigm4.com/HTMLmanual/15.7/scidb_ug/) that is required for such a feature rich system. The documentation is however lacking in clarity around loading data with most examples being based around the assumption that the data already exists within SciDB or is being generated by SciDB. While installation on a single server is relatively straight forward (for older versions with binaries supplied for supported platforms) the process is hard to identify as the community edition installation documentation is mixed in with the documentation on installation of the enterprise edition of SciDB. Access to source code is via tarballs; there is no source control system with general access to investigate the history of the project in detail.

## MOTIVATION

PhilDB aims at handling data for exploratory purposes with the intention to later integrate with other systems, with minimal initial deployment overhead. It is assumed that the smaller time series database systems discussed previously derive from similar needs. It has been found "[m]ost scientists are adamant about not discarding any data" (*Cudré-Mauroux et al., 2009*). In particular, experience in hydrology has found that hydrological data requires the ability to track changes to it, since streamflow discharge can be regularly updated through quality control processes or updates to the rating curves used to convert from water level to discharge. Open-source 'big data' time series database offerings don't support the ability to track any changed values out of the box (such support would have to be developed external to the system). Their design targets maximum efficiency of write-once and read-many operations. When streamflow data is used within forecasting systems, changes to the data can alter the forecast results. Being able to easily identify if a change in forecast results is due to data or code changes greatly simplifies resolving issues

during development and testing. Therefore, both requirements of minimal deployment overhead and logging of any changed values rule out the current 'big data' systems.

While SciDB does address the data tracking need, recent versions of the community edition are complex to install since they require building from source, a process more involved than the usual './configure; make; make install'. Older versions are more readily installed on supported platforms, however the system is still complex to use, requires root access to install, a working installation of PostgreSQL and a dedicated user account for running. Installation difficulty isn't enough to rule out the system being a suitable solution, but it does diminish its value as an exploratory tool. SciDB is also licensed under the GNU Affero General Public License (AGPL), that can be perceived as a problem in corporate or government development environments. In these environments, integration with more liberally licensed (e.g. Apache License 2.0 or 3-clause BSD) libraries is generally preferred with many online discussions around the choice of liberal licences for software in the scientific computing space. For example, it can be argued that a simple liberal license like the BSD license encourages the most participation and reuse of code (*Brown, 2015*; *VanderPlas, 2014*; *Hunter, 2004*).

Finally, SciDB has a broader scope than just storage and retrieval of time series data, since "SciDB supports both a functional and a SQL-like query language" (*Stonebraker et al., 2011*). Having SQL-like query languanges does allow for SciDB to readily support many high performance operations directly when handling large already loaded data. These query languages do, however, add additional cognitive load (*Sweller, Ayres & Kalyuga, 2011*) for any developer interfacing with the system as the query languages are specific to SciDB. If using SciDB for performing complex operations on very large multi-dimensional array datasets entirely within SciDB, learning these query languages would be well worth the time. The Python API does enable a certain level of abstraction between getting data out of SciDB and into the scientific Python ecosystem.

Of the other existing systems discussed here, none support logging of changed values. Limited documentation makes them difficult to evaluate, but from what can be seen and inferred from available information, the designs are targeted at the 'write once, read many' style of the 'big data' time series systems at a smaller deployment scale. These systems were extremely early in development or yet to be started at time work began on PhilDB in October 2013.

The need to be fulfilled is purely to store time series of floating point values and extract them again for processing with other systems.

## Use case

To summarise, PhilDB has been created to provide a time series database system that is easily deployed, used, and has logging features to track any new or changed values. It has a simple API for writing both new and updated data with minimal user intervention. This is to allow for revising time series from external sources where the data can change over time, such as streamflow discharge data from water agencies. Furthermore, the simple API extends to reading, to enable easy retrieval of time series, including the ability to read time series as they appeared at a point in time from the logs.

# ARCHITECTURE

PhilDB uses a central 'meta-data store' to track the meta information about time series instances. Relational databases are a robust and reliable way to hold related facts. Since the meta data is simply a collection of related facts about a time series, a relational database is used for the meta-data store. Time series instances are associated with a user chosen identifier and attributes and each time series instance is assigned a UUID (*Leach, Mealling & Salz, 2005*) upon creation, all of which is stored in the meta-data store. The actual time series data (and corresponding log) is stored on disk with filenames based on the UUID (details of the format are discussed in the 'Database Format' section). Information kept in the meta-data store can then be used to look up the UUID assigned to a given time series instance based on the requested identifier and attributes. Once the UUID has been retrieved, accessing the time series data is a simple matter of reading the file from disk based on the expected UUID derived filename.

## Architecture philosophy

The reasoning behind this architectural design is so that:

* An easy to use write method can handle both new and updated data (at the same time if needed).
* Read access is fast and easy for stored time series.
* Time series are easily read as they appeared at a point in time.
* Each time series instance can be stored with minimal initial effort.

Ease of writing data can come at the expense of efficiency to ensure that create, update or append operations can be performed with confidence that any changes are logged without having to make decisions on which portions of the data are current or new. The expectation is that read performance has a greater impact on use as they are more frequent. Attaching a time series identifier as the initial minimal information allows for data from a basic dataset to be loaded and explored immediately. Additional attributes can be attached to a time series instance to further differentiate datasets that share conceptual time series identifiers. By default, these identifier and attribute combinations are then stored in a tightly linked relational database. Conceptually this meta-data store could optionally be replaced by alternative technology, such as flat files. As the data is stored in individual structured files, the meta-data store acts as a minimal index with most of the work being delegated to the operating system and in turn the file system.

# IMPLEMENTATION

PhilDB is written in Python because it fits well with the scientific computing ecosystem (*Perez, Granger & Hunter, 2011*). The core of the PhilDB package is the PhilDB database class (http://phildb.readthedocs.org/en/latest/api/phildb.html#module-phildb.database), that exposes high level methods for data operations. These high level functions are designed to be easily used interactively in the IPython interpreter (*Perez & Granger, 2007*) yet still work well in scripts and applications. The goal of interactivity and scriptability are to enable

exploratory work and the ability to automate repeated tasks (*Shin et al., 2011*). Utilising Pandas (*McKinney, 2012*) to handle complex time series operations simplifies the internal code that determines if values require creation or updating. Returning Pandas objects from the read methods allows for data analysis to be performed readily without further data munging. Lower level functions are broken up into separate modules for major components such as reading, writing, and logging, that can be easily tested as individual components. The PhilDB class pulls together the low level methods, allowing for the presentation of a stable interface that abstracts away the hard work of ensuring that new or changed values, and only those values, are logged.

Installation of PhilDB is performed easily within the Python ecosystem using the standard Python setup.py process, including installation from PyPI using 'pip.'

## Features

Key features of PhilDB are:

* A single write method accepting a pandas.Series object, data frequency and attributes for writing or updating a time series.
* A read method for reading a single time series based on requested time series identifier, frequency and attributes.
* Advanced read methods for reading collections of time series.
* Support for storing regular and irregular time series.
* Logging of any new or changed values.
* Log read method to extract a time series as it appeared on a given date.

## Database format

The technical implementation of the database format, as implemented in version 0.6.1 of PhilDB (*MacDonald, 2015*), is described in this section. Due to the fact that PhilDB is still in the alpha stage of development the specifics here may change significantly in the future.

The meta-data store tracks attributes using a relational database, with the current implementation using SQLite (*Hipp, Kennedy & Mistachkin, 2015*). Actual time series data are stored as flat files on disk, indexed by the meta-data store to determine the path to a given series. The flat files are implemented as plain binary files that store a 'long,' 'double,' and 'int' for each record. The 'long' is the datetime stored as a 'proleptic Gregorian ordinal' as determined by the Python datetime.datetime.toordinal method (https://docs.python.org/2/library/datetime.html#datetime.date.toordinal) (*Van Rossum, 2015*). The 'double' stores the actual value corresponding to the datetime stored in the preceding 'long'. Finally, the 'int' is a meta value for marking additional information about the record. In this version of PhilDB the meta value is only used to flag missing data values. Individual changes to time series values are logged to HDF5 files (*HDF Group, 1997*) that are kept alongside the main time series data file with every new value written as a row in a table, each row having a column to store the date, value, and meta value as per the file format. In addition, a final column is included to record the date and time the record was written.

**Table 1** **Breakdown of length of time series in the evaluation dataset (all values rounded to nearest day).**

| Mean | 16,310 days |
| --- | --- |
| Std | 2,945 days |
| Min | 10,196 days |
| 25% | 14,120 days |
| 50% | 15,604 days |
| 75% | 18,256 days |
| Max | 22,631 days |

## Distribution of lengths of time series

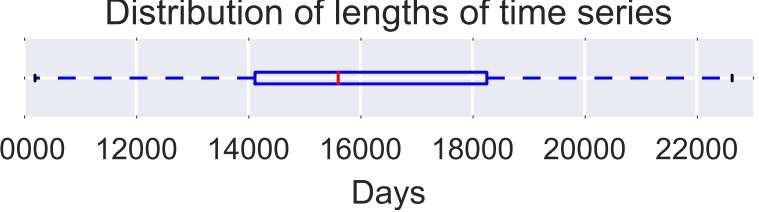

**Figure 1** **Distribution of time series length for the 221 time series in the evaluation dataset.**

# EVALUATION

Of the open-source systems evaluated (as identified in the section 'Open-source Systems'), InfluxDB came the closest in terms of minimal initial installation requirements and feature completeness, however, it doesn't support the key feature of update logging. Contrasting with InfluxDB, SciDB met the requirement of time series storage with update logging but didn't meet the requirement for simplicity to deploy and use. Both these systems were evaluated in comparison to PhilDB.

## Evaluation dataset

The Hydrological Reference Stations (*Zhang et al., 2014*) dataset from the Australian Bureau of Meteorology (http://www.bom.gov.au/water/hrs/) was used for the evaluation. This dataset consists of daily streamflow data for 221 time series with a mean length of 16,310 days, the breakdown of the series lengths are in Table 1 and visualised in Fig. 1.

## Evaluation method

Three key aspects were measured during the evaluation:

∗ Write performance
∗ Read performance
∗ Disk usage

Ease of installation and use, while subjective, is also discussed in the installation and usage sections related to each database.

To simplify the evaluation process and make it easily repeatable, the SciDB 14.3 virtual appliance image (SciDB14.3-CentOS6-VirtualBox-4.2.10.ova from https://downloads.paradigm4.com/) was used to enable easy use of the SciDB database. This virtual appliance

was based on a CentOS Linux 6.5 install. The PhilDB and InfluxDB databases were installed into the same virtual machine to enable comparison between systems. The virtual machine host was a Mid-2013 Apple Macbook Air, with a 1.7 GHz Intel Core i7 CPU, 8GB of DDR3 RAM and a 500GB SSD hard drive. VirtualBox 4.3.6 r91406 was used on the host machine for running the virtual appliance image with the guest virtual machine being allocated 2 processors and 4GB of RAM.

Write performance was evaluated by writing all time series from the evaluation dataset (described in the section 'Evaluation Dataset') into the time series databases being evaluated. This first write will be referred to as the initial write for each database. To track the performance of subsequent updates and reading the corresponding logged time series a further four writes were performed. These writes will be referred to as 'first update' through to 'fourth update'. The update data was created by multiplying some or all of the original time series by 1.1 as follows:

* First update: multiplied the last 10 values in the time series by 1.1 leaving the rest of the record the same.
* Second update: multiplied the first 10 values by 1.1, resulting in reverting the previously modified 10 values.
* Third update: multiplied the entire original series by 1.1 resulting in an update to all values aside from the first 10.
* Fourth update: the original series multiplied by 1.1 again, which should result in zero updates.

The SciDB load method used in this experiment did not support updating individual values. The entire time series needed to be passed or the resulting array would consist of only the supplied values. Due to this only full updates were tested and not individual record updates or appends.

Performance reading the data back out of each database system was measured by recording the time taken to read each individual time series, after each update, and analysing those results.

As can be seen by Fig. 2, InfluxDB performance was a long way behind SciDB and PhilDB. Given the performance difference and that InfluxDB doesn't support change logging only the initial load and first read were performed for InfluxDB.

Disk usage was measured by recording the size of the data directories as reported by the 'du' Unix command. The size of the data directory was measured before loading any data and subtracted from subsequent sizes. Between each data write (initial load and four updates) the disk size was measured to note the incremental changes.

For both PhilDB and SciDB the evaluation process described in this section was performed four times and the mean of the results analysed. Results between the four runs were quite similar so taking the mean gave results similar to the individual runs. Analysing and visualising an individual run rather than the mean would result in the same conclusions.

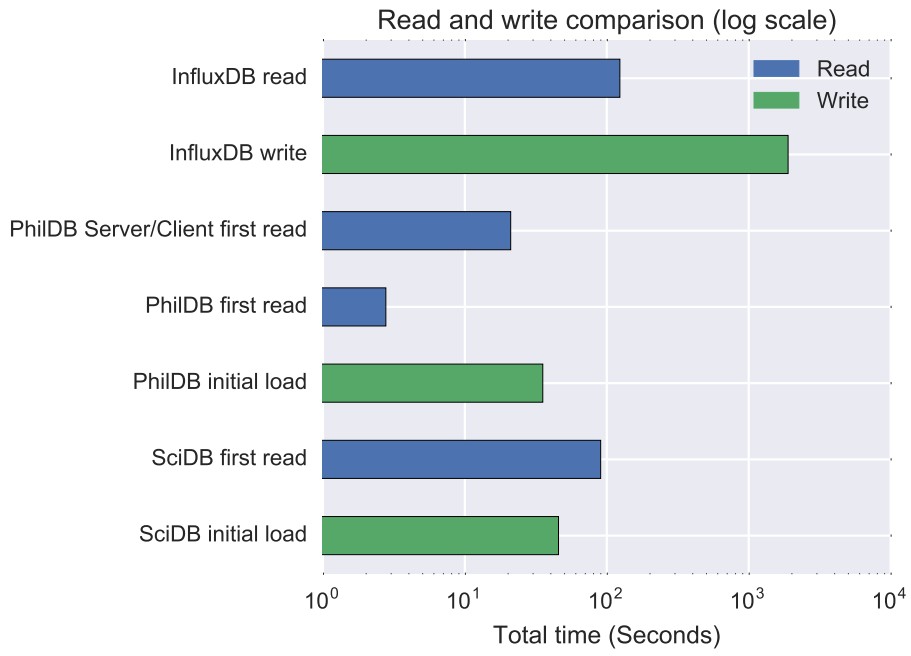

**Figure 2** **Total write/read time for 221 daily time series.**

## Evaluated databases

This section discusses each of the evaluated databases. Firstly, they are introduced and then their installation and usage is considered.

### *InfluxDB*

Paul Dix (CEO of InfluxDB) found that performance and ease of installation were the main concerns of users of existing open-source time series database systems (*Dix, 2014*). InfluxDB was built to alleviate both those concerns.

While InfluxDB is designed for high performance data collection, it is not designed for bulk loading of data. Searching the InfluxDB issue tracker on GitHub (https: //github.com/influxdata/influxdb/issues), it can be seen that bulk loading has been a recurring problem with improvement over time. Bulk loading performance is, however, still poor compared to SciDB and PhilDB, as seen later in the performance results (section 'Performance'). A key feature of interest with InfluxDB was the ability to identify time series with tags. This feature is in line with the attributes concept used by PhilDB, thereby allowing multiple time series to be grouped by a single key identifier but separated by additional attributes or tags.

**Installation:** InfluxDB is easily installed compared to the other open-source systems reviewed, as demonstrated by the short install process shown below. Installation of pre-built packages on Linux requires root access (https://influxdb.com/docs/v0.9/introduction/ installation.html). Installation of InfluxDB was performed in the CentOS Linux 6.5 based virtual machine containing the pre-installed SciDB instance.

```
wget http://influxdb.s3.amazonaws.com/influxdb-0.9.6.1-1.x86_64.rpm
sudo yum localinstall influxdb-0.9.6.1-1.x86_64.rpm
```

Starting the InfluxDB service with:

```
sudo /etc/init.d/influxdb start
```

**Usage:** Loading of data into the InfluxDB instance was performed using the InfluxDB Python API that was straight forward to use. However, poor performance of bulk loads lead to a lot of experimentation on how to most effectively load large amounts of data quickly, including trying curl and the Influx line protocol format directly. The final solution used was to chunk the data into batches of 10 points using the Pandas groupby functionality before writing into InfluxDB using the InfluxDB Python API DataFrameClient write_points method, for example:

```
streamflow = pandas.read_csv(filename, parse_dates=True, index_col=0, header = None)
for k, g in streamflow.groupby(np.arange(len(streamflow))//100):
    influx_client.write_points(g, station_id)
```

In addition to experimenting with various API calls, configuration changes were attempted resulting in performance gains by lowering values related to the WAL options (the idea was based on an older GitHub issue discussing batch loading (https://github.com/influxdata/influxdb/issues/3282) and WAL tuning to improve performance). Despite all this effort, bulk data loading with InfluxDB was impractically slow with a run time generally in excess of one hour to load the 221 time series (compared to the less than 2 minutes for SciDB and PhilDB). Reading was performed using the Python API InfluxDBClient query method:

```
streamflow = influx_client.query('SELECT_*_FROM_Q{0}'.format('410730'))
```

### PhilDB

PhilDB has been designed with a particular use case in mind as described in the 'Use Case' section. Installation of PhilDB is quite easy where a compatible Python environment exists. Using a Python virtualenv removes the need to have root privileges to install PhilDB and no dedicated user accounts are required to run or use PhilDB. A PhilDB database can be written to any location the user has write access, allowing for experimentation without having to request a database be created or needing to share a centralised install.

**Installation:** Installation of PhilDB is readily performed using pip:

```
pip install phildb
```

**Usage:** The experimental dataset was loaded into a PhilDB instance using a Python script. Using PhilDB to load data can be broken into three key steps.

First, initialise basic meta information:

```
db.add_measurand('Q', 'STREAMFLOW', 'Streamflow')
db.add_source('BOM_HRS', 'Bureau_of_Meteorology;_Hydrological_Reference_Stations_
    dataset.')
```

This step only need to be performed once, when configuring attributes for the PhilDB instance for the first time, noting additional attributes can be added later.

Second, add an identifier for a time series and a time series instance record based on the identifier and meta information:

```
db.add_timeseries(station_id)
db.add_timeseries_instance(station_id, 'D', '', measurand = 'Q', source = 'BOM_HRS')
```

Multiple time series instances, based on different combinations of attributes, can be associated with an existing time series identifier. Once a time series instance has been created it can be written to and read from.

Third, load the data from a Pandas time series:

```
streamflow = pandas.read_csv(filename, parse_dates=True, index_col=0, header = None)
db.write(station_id, 'D', streamflow, measurand = 'Q', source = 'BOM_HRS')
```

In this example the Pandas time series is acquired by reading a CSV file using the Pandas read_csv method, but any data acquisition method that forms a Pandas.Series object could be used. Reading a time series instance back out is easily performed with the read method:

```
streamflow = db.read(station_id, 'D', measurand = 'Q', source = 'BOM_HRS')
```

The keyword arguments are optional provided the time series instance can be uniquely identified.

### SciDB

SciDB, as implied by the name, was designed with scientific data in mind. As a result SciDB has the feature of change logging, allowing past versions of series to be retrieved. Unfortunately SciDB only identifies time series by a single string identifier, therefore storing multiple related time series would require externally managed details about what time series are stored and with what identifier. Due to the sophistication of the SciDB system it is relatively complex to use with two built in languages, AFL and AQL, that allow for two different approaches to performing database operations. This, in turn, increases the amount of documentation that needs to be read to identify which method to use for a given task (such as writing a time series into the database). While the documentation is comprehensive in detailing the available operations, it is largely based on the assumption that the data is already within SciDB and will only be operated on within SciDB, with limited examples on how to load or extract data via external systems.

**Installation:** SciDB does not come with binary installers for newer versions and the build process is quite involved. Instructions for the build proccess are only available from the SciDB forums using a registered account (http://forum.paradigm4.com/t/release-15-7/843). Installation of older versions is comparable to InfluxDB with the following steps listed in the user guide:

```
yum install -y https://downloads.paradigm4.com/scidb-14.12-repository.rpm
yum install -y scidb-14.12-installer
```

Same as InfluxDB, SciDB requires root access to install and a dedicated user account for running the database. A PostgreSQL installation is also required by SciDB for storing information about the time series data that SciDB stores. Unlike InfluxDB, SciDB has authentication systems turned on by default that requires using dedicated accounts even for basic testing and evaluation.

Only Ubuntu and CentOS/RHEL Linux variants are listed as supported platforms in the install guide.

**Usage:** It took a considerable amount of time to identify the best way to load data into a SciDB instance, however once that was worked out, the actual load was quick and effective consisting of two main steps.

First, a time series needs to be created:

```
iquery -q "CREATE_ARRAY_Q${station}_<date:datetime,_streamflow:double>_[i
    =0:*,10000,0];"
```

It is worth noting that datetime and double need to be specified for time series storage, since SciDB can hold many different array types aside from plain time series. Additionally, SciDB identifiers can not start with a numeric character so all time series identifiers were prefixed with a 'Q' (where 'Q' was chosen in this case because it is conventionally used in the hydrological context to represent streamflow discharge).

Second, the data is written using the iquery LOAD method as follows:

```
iquery -n -q "LOAD_Q${station}_FROM_'/home/scidb/${station}.scidb';"
```

This method required creating data files in a specific SciDB text format before using the csv2scidb command that ships with SciDB.

Identifying the correct code to read data back out required extensive review of the documentation, but was quick and effective once the correct code to execute was identified. The SciDB Python code to read a time series back as a Pandas.DataFrame object is as follows:

```
streamflow = sdb.wrap_array('Q' + station_id).todataframe()
```

A contributing factor to the difficulty of identifying the correct code is that syntax errors with the AQL based queries (using the SciDB iquery command or via the Python API) are at times uninformative about the exact portion of the query that is in error.

## Performance

It should be noted that PhilDB currently only supports local write, which is advantageous for performance, compared to InfluxDB that only supports network access. InfluxDB was hosted locally, which prevents network lag, but the protocol design still reduced performance compared to the direct write as done by PhilDB. Although SciDB has network access, only local write performance (using the SciDB iquery command) and network based read access (using the Python API) were evaluated. SciDB was also accessed locally to avoid network lag when testing the network based API. For a comparable network read access comparison the experimental PhilDB Client/Server software was also used.

### *Write performance*

Write performance was measured by writing each of the 221 time series into the database under test and recording the time spent per time series.

As can be seen in Fig. 2, SciDB and PhilDB have a significant performance advantage over InfluxDB for bulk loading of time series data. SciDB write performance is comparable to PhilDB, so a closer comparison between just SciDB and PhilDB write performance is shown in Fig. 3.

It can be seen that while PhilDB has at times slightly better write performance, SciDB has more reliable write performance with a tighter distribution of write times. It can also be seen from Fig. 3 that write performance for SciDB does marginally decrease as more updates are written. PhilDB write performance while more variable across the dataset is also variable in performance based on how much of the series required updating. Where the

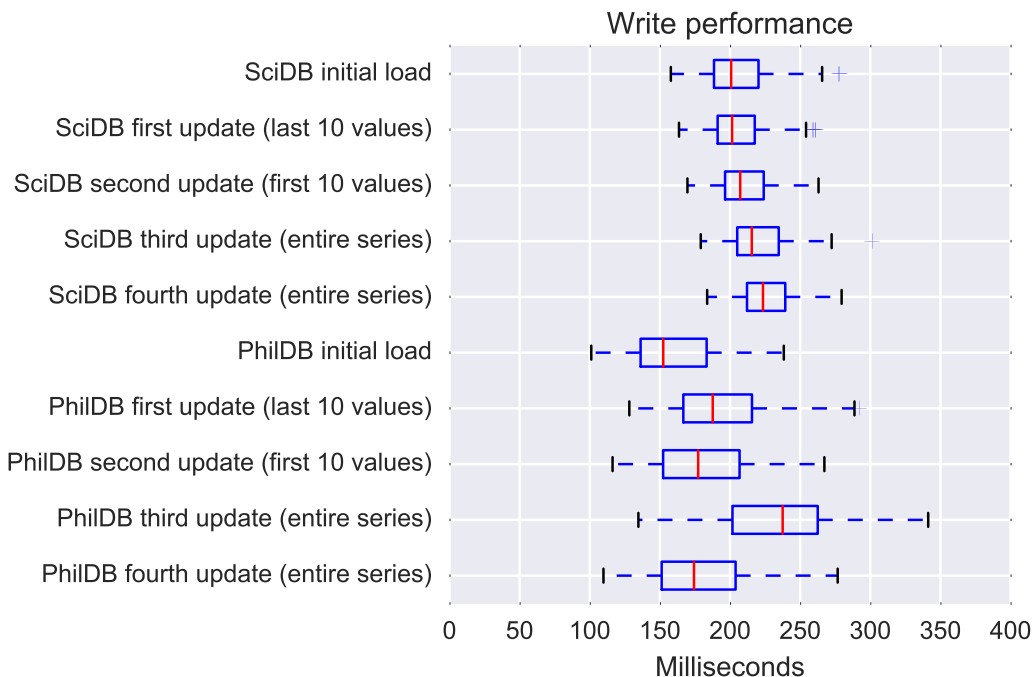

**Figure 3** Distribution of write times for 221 time series.

fourth update writes the same data as the third update it can be seen that the performance distribution is closer to that of the initial load than the third load, since the data has actually remained unchanged.

Both SciDB and PhilDB perform well at loading datasets of this size with good write performance.

### Read performance

InfluxDB read performance is adequate and SciDB read speed is quite good, however PhilDB significantly out-performs both InfluxDB and SciDB in read speed, as can be seen in Fig. 2. Even the PhilDB server/client model, which has yet to be optimised for performance, out-performed both InfluxDB and SciDB. Read performance with PhilDB is consistent as the time series are updated, as shown in Fig. 4, due to the architecture keeping the latest version of time series in a single file. Reading from the log with PhilDB does show a decrease in performance as the size of the log grows, but not as quickly as SciDB. While PhilDB maintains consistent read performance and decreasing log read performance, SciDB consistently decreases in performance with each update for reading both current and logged time series.

### Disk usage

After the initial load InfluxDB was using 357.21 megabytes of space. This may be due to the indexing across multiple attributes to allow for querying and aggregating multiple time series based on specified attributes. This is quite a lot of disk space being used compared to SciDB (93.64 megabytes) and PhilDB (160.77 megabytes) after the initial load. As can be

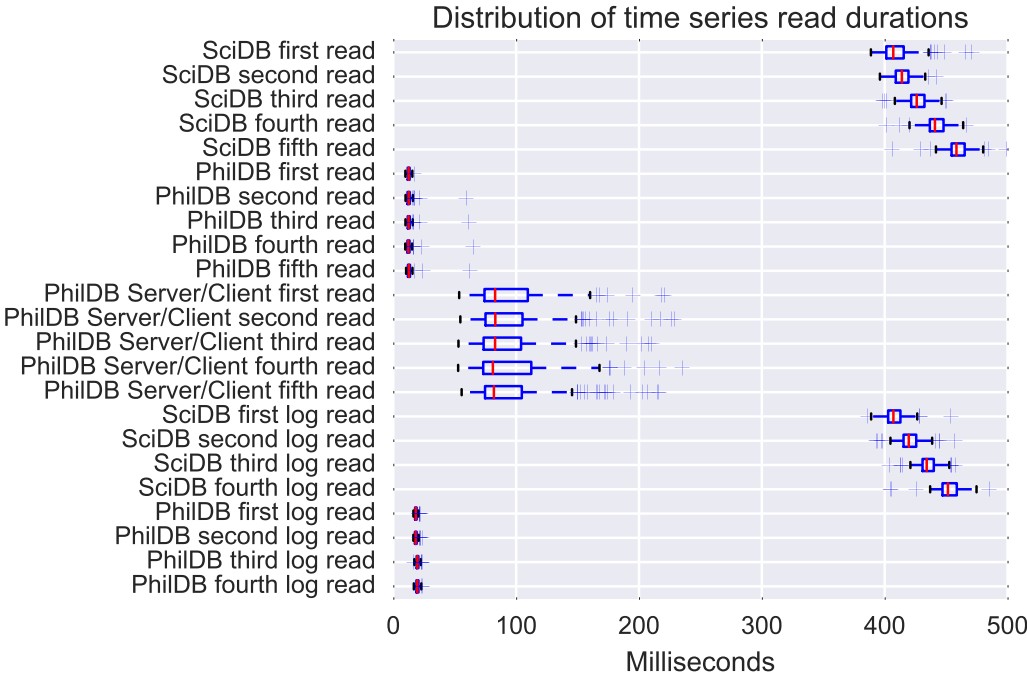

**Figure 4** **Distribution of read durations for the 221 time series from the evaluation dataset.**

seen in Fig. 5, SciDB disk usage increases linearly with each update when writing the entire series each time. In contrast, updates with PhilDB only result in moderate increases and depends on how many values are changed. If the time series passed to PhilDB for writing is the same as the already stored time series then no changes are made and the database size remains the same, as can be seen between update 3 and 4 in Fig. 5.

### Performance summary

Each database has different design goals that results in different performance profiles. InfluxDB is not well suited to this use case with a design focusing on high performance writing of few values across many time series for metric collection, leading to poor performance for bulk loading of individual time series.

SciDB fares much better with consistent read and write performance, with slight performance decreases as time series are updated, likely due to design decisions that focus on handling large multi-dimensional array data for high performance operations. Design decisions for SciDB that lead to consistent read and write performance appear to also give the same read performance when accessing historical versions of time series. Achieving consistent read and write performance (including reading historical time series) seems to have come at the expense of disk space with SciDB consuming more space than PhilDB and increasing linearly as time series are updated.

PhilDB performs quite well for this particular use case, with consistently fast reads of the latest time series. This consistent read performance does come at the expense of reading historical time series from the logs, which does degrade as the logs grow. Write performance for PhilDB, while variable, varies due to the volume of data changing.

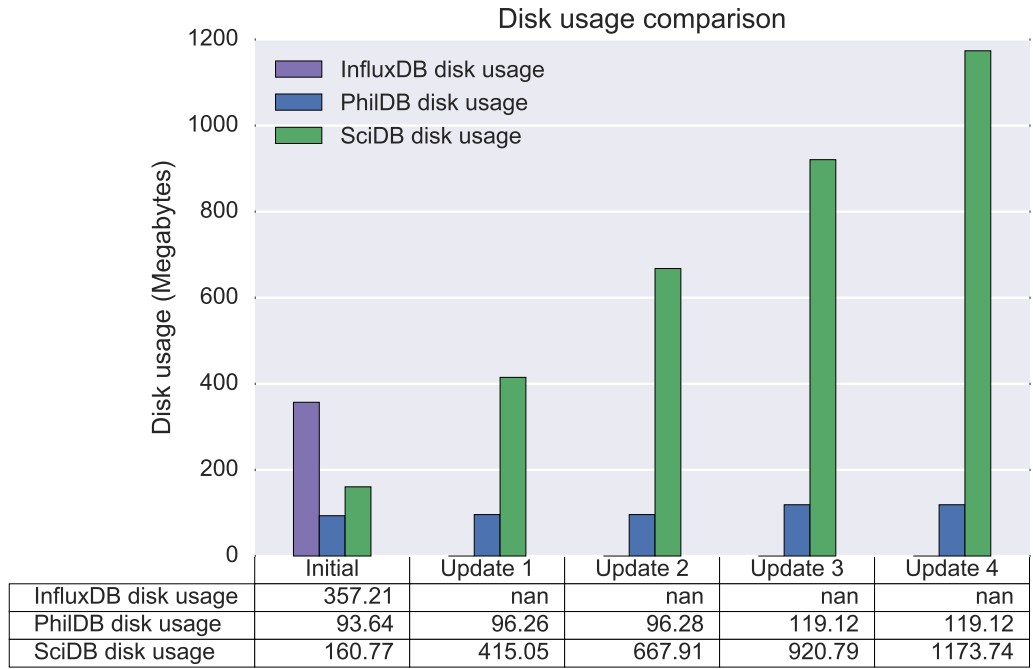

| | Initial | Update 1 | Update 2 | Update 3 | Update 4 |
|---|---|---|---|---|---|
| InfluxDB disk usage | 357.21 | nan | nan | nan | nan |
| PhilDB disk usage | 93.64 | 96.26 | 96.28 | 119.12 | 119.12 |
| SciDB disk usage | 160.77 | 415.05 | 667.91 | 920.79 | 1173.74 |

**Figure 5** Disk usage after initial data load and each subsequent data update.

The performance of PhilDB (particularly the excellent read performance) compared to SciDB for this use case was unexpected since the design aimed for an easy to use API at the expense of efficiency.

## FUTURE WORK

PhilDB is still in its alpha stage. Before reaching the beta stage, development efforts shall investigate:

* Complete attribute management to support true arbitrary attribute creation and attachment.
* Possible alternative back ends, using alternative data formats, disk paths, and relational databases.
* More sophisticated handling of time zone meta-data.
* Storage of quality codes or other row level attributes.
* Formalisation of UUID usage for sharing of data.

## CONCLUSION

In conclusion, there is a need for an accessible time series database that can be deployed quickly so that curious minds, such as those in our scientific community, can easily analyse time series data and elucidate world-changing information. For scientific computing, it is important that any solution is capable of tracking subsequent data changes.

Although InfluxDB comes close with features like tagging of attributes and a clear API, it lacks the needed change logging feature and presently suffers poor performance for bulk

loading of historical data. InfluxDB has clearly been designed with real-time metrics based time series in mind and as such doesn't quite fit the requirements outlined in this paper.

While SciDB has the important feature of change logging and performs quite well, it doesn't have a simple mechanism for tracking time series by attributes. SciDB is well suited for handing very large multi-dimensional arrays, which can justify the steep learning curve for such work, but for input/output of plain time series such complexity is a little unnecessary.

PhilDB addresses this gap in existing solutions, as well as surpassing them for efficiency and usability. Finally, PhilDB's source code has been released on GitHub (https://github.com/amacd31/phildb) under the permissive 3-clause BSD open-source license to help others easily extract wisdom from their data.

## ACKNOWLEDGEMENTS

I would like to thank Di MacDonald for her editorial advice on various drafts, my fiancée Katrina Cornelly for her support and editorial advice, and my colleague Richard Laugesen for his valuable review comments on an earlier draft. PhilDB was named in memory of my father Phillip MacDonald.

### Funding
The author received no funding for this work.

### Competing Interests
The author declares there are no competing interests.

### Author Contributions
- Andrew MacDonald conceived and designed the experiments, performed the experiments, analyzed the data, wrote the paper, prepared figures and/or tables, performed the computation work, reviewed drafts of the paper.

### Data Availability
Source code stored on Zenodo (zenodo.org) with the DOI 10.5281/zenodo.32437 accessible at 10.5281/zenodo.32437 or https://zenodo.org/record/32437.

### Supplemental Information
Supplemental information for this article can be found online at http://dx.doi.org/10.7717/peerj-cs.52#supplemental-information.

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
