# Peer review of "PhilDB: the time series database with built-in change logging"

_PeerJ Computer Science, doi:10.7717/peerj-cs.52_

## Round 0.1 · original submission · Major Revisions

Please follow all the reviewers' recommendations. Pay attention specially to English grammar. If possible, ask help to a professional English proofreader.

Reviewer 1 ·

Basic reporting

The document is written in a clear and concise way, perhaps a little shorter than usual. It seems to be written as a report instead of an article. Motivation should be extended and its proposal better justified. I consider that the complexity of the installation of a database system is not reason enough to discard it. It will be one of the many other features to take into consideration. In fact, this list should be added and used to compare PhilDB with other systems in the experimentation section. The organization of the paper is clear and correct but the experimentation should be extended and improved.

Experimental design

Regarding the experimentation, Phildb is only compared to InfluxDB because it is the closest in terms of minimal installation requirements and feature completeness but the paper also mentions SciDB as the only open source solution that addresses the scientific need to track changes to stored time data. Why is SciDB not evaluated in the experiment? Besides, this provides a SQL-like query language which has been always welcome and necessary in order to make the use of these systems easier. Datasets used should be better described and be available; alternatively datasets used in other benchmarks could be also used. The different metrics to be evaluated and the method to do it should be mentioned previously and then discuss the results. The paper does not show clearly how the advantage of including a built-in change logging could affect to other performance parameters of the system. In short, the benchmark should be carefully planned and run in several time series databases and be reproducible in order to community considers its results valid

Validity of the findings

This issue must be improved as mentioned in the previous section.

Reviewer 2 ·

Basic reporting

The sample dataset used in experiments have not been submitted nor cited as required by journal policy.

The language may be improved, as there are some long sentences difficult to read. For example, "...easier to integrate with (compared with proprietary systems) and they are more fitting..." Isn't it better with the included parenthesis or other phrase structure? By the way, "are more fitting" -> "fit better"?

In my humble opinion, the novel system proposed architecture is weakly explained. Several ideas to improve: explain all relevant details, maybe some figures can clarify, what about the use cases in which the log is useful (specially in research).

Experimental design

The paper could benefit from a more rigorous experimentation. Some ideas:
- Comparing with the only other TS DB mentioned with logging (Stonebraker).
- Would not data from table 1 be better represented in a quartile plot (with a note including mean and std)?
- What about performing experiments on the use of the logging facility?
- The comment about the problem with dates in InfluxDB deserves a bit of work to actually explain why InfluxDB can not be used. Being open source the code is available and in a time series database this should be something that can be answered.
- The results about the space required by each database could be commented explaining which are the differences in the storage mechanism. is there any functionality in InfluxDB not supported in PhilDB. This may be ok. It is just that it would be good to know when each one fits better.

Validity of the findings

No comments

Additional comments

Interesting idea. It seems a good piece of software. However I think the paper needs revision to fit in a research journal.

---

## Round 0.2 · Minor Revisions

Reviewer 1 has still some concerns about the writing style of the paper. Please prepare a new review taking in to account his previous suggestions.

Reviewer 1 ·

Basic reporting

The author has taken into account the suggestions proposed in the first review, except the style of writing. It is written as a report. Introduction should have the context of the problem, motivation, the goal of the paper and how it is organised, then, background, architecture, implementation and evaluation. In research paper sentences such as “The author’s interest is derived …” or “The need of the author is …” must be avoided, it’s better to write “Phildb aims at handling data for exploratory purposes….”
The adjective “simple” in the last sentence of the summary devalues the paper.
Evaluation section should be better organised, it would be good to add a paragraph or table where you indicate the metrics to be evaluated before starting to show results. Dataset description should be described before the experiment as well as the methodology followed; later, results and discussion. Sections 6.2,6.3 and 6.4 should be grouped in one subsection since it exposes DB setting preliminaries or moved to an addendum since their description is this section breaks the threat of the evaluation.

Experimental design

As previously said, evaluation section should be rewritten. Now the experimentation is acceptable although the evaluation of another dataset with different features would be welcome.

Validity of the findings

correct

Additional comments

I consider this database tries to fill a gap in the temporal series database market and thus this contribution is valuable. The style of writing must be enhanced.

Reviewer 2 ·

Basic reporting

All basic issues commented have been reasonably addressed and the paper has improved.

Experimental design

I think the experimental design is clearer, reproducible and reasonably complete now.

Validity of the findings

Experiment results are meaningful now.

Additional comments

No comments.

---

## Round 0.3 · accepted · Accept

Author followed the recommendations of reviewers and the paper has considerably improved, so I think it is ready for publication. Congratulations.